# Stereoscopic Depth Perception and Visuospatial Dysfunction in Alzheimer’s Disease

**DOI:** 10.3390/healthcare9020157

**Published:** 2021-02-03

**Authors:** Nam-Gyoon Kim, Ho-Won Lee

**Affiliations:** 1Department of Psychology, Keimyung University, Daegu 42601, Korea; 2Department of Neurology, School of Medicine & Brain Science and Engineering Institute, Kyungpook National University, Daegu 41566, Korea; neuromd@knu.ac.kr

**Keywords:** Alzheimer’s disease (AD), mild cognitive impairment (MCI), visuospatial dysfunction, biomarker, stereopsis, binocular disparity, coarse disparity, preclinical AD, middle temporal area (MT)

## Abstract

With visuospatial dysfunction emerging as a potential marker that can detect Alzheimer’s disease (AD) even in its earliest stages and with disturbance in stereopsis suspected to be the prime contributor to visuospatial deficits in AD, we assessed stereoscopic abilities of patients with AD and mild cognitive impairment (MCI). Whereas previous research assessing patients’ stereoacuity has yielded mixed results, we assessed patients’ capacity to process coarse disparities that can convey adequate depth information about objects in the environment. We produced two virtual cubes at two different distances from the observer by manipulating disparity type (absolute vs. relative), disparity direction (crossed vs. uncrossed) and disparity magnitude, then had participants judge the object that appeared closer to them. Two patient groups performed as well as, or even better than elderly controls, suggesting that AD patients’ coarse disparity processing capacity is capable of supporting common tasks involving reaching, grasping, driving, and navigation. Results may help researchers narrow down the exact cause(s) of visuospatial deficits in AD and develop and validate measures to assess visuospatial dysfunction in clinical trials and disease diagnosis.

## 1. Introduction

Alzheimer’s disease (AD) is the most common cause of dementia, accounting for 60–70% of all cases worldwide. As a degenerative disorder, AD is characterized by progressive neuronal loss in the central nervous system, which leads to deficits in memory and cognitive skills. Pathologically, the accumulation of two types of abnormal protein aggregates (extracellular senile plaques (SPs) composed of amyloid-beta (Aβ) protein and intracellular neurofibrillary tangles (NFTs) composed of hyperphosphorylated tau protein), is suspected to drive neuronal degeneration and eventual cell death in AD [1]. Despite intensive research over the past three decades, there is still no effective treatment to stop or slow the progression of the disease. However, medication is available that can temporarily relieve some of the symptoms in some people.

At present, AD is primarily diagnosed based on clinical and neuropsychological evaluations complemented by brain imaging studies. Accuracy of a clinical diagnosis, however, is rather poor with accuracy rate ranging between 65% and 92% [2,3,4,5]. A definitive diagnosis of AD can only be made at autopsy by the detection of two hallmark pathologies, i.e., SPs and NFTs. Notably, autopsy studies have revealed these pathological signatures even in the brains of cognitively intact individuals. Accumulation of AD pathology is shown to correlate highly with symptoms of AD type dementia [5,6]. Thus, pathological lesions present in cognitively normal individuals suggest that neuropathological processes of AD have begun years before clinical symptoms become apparent [7]. 

### 1.1. Preclinical AD

Recent advances in brain imaging techniques and new methods to analyze cerebrospinal fluid (CSF) have enabled researchers to develop several biomarkers for the diagnosis of AD. In numerous clinical trials, these biomarkers have demonstrated consistent results reflecting the core pathology of the disease, thereby improving diagnostic criteria for AD. Significantly, these biomarkers, when used in combination, are capable of predicting cognitive decline even in cognitively intact individuals [8,9]. Based on the evidence accumulated to date, AD is now recognized as a continuous process that begins 15 to 20 years before clinical symptoms emerge [10,11,12,13,14]. This long period is referred to as "preclinical" to classify individuals with normal cognition but at risk of developing AD based on biomarker evidence of accumulating AD pathology.

Almost all symptom-modifying drugs for AD developed recently have proven to be ineffective in clinical trials [7,15,16]. Perhaps these trials were initiated when the neurodegeneration process had advanced to the irreversible stage. Treatments could have been more effective if administered early in the disease process before symptoms emerge. Thus, it is critical to be able to detect individuals who are likely to develop AD in their earliest preclinical stages to facilitate early implementation of therapeutic interventions and to enhance chances of delaying or even preventing debilitating neural deterioration. Combined use of CSF and imaging biomarkers has been shown effective in providing diagnostically reliable information about AD pathology in the earliest stages of the disease. However, current biomarker tests are expensive (PET scan), invasive (lumbar puncture to obtain CSF samples), and less accessible (PET scan), making them unsuitable for routine clinical practice. There is, therefore, a pressing need to develop easily accessible, cost-effective and non-invasive biomarkers for early detection of AD.

### 1.2. Visuospatial Dysfunction as a Biomarker for AD

AD has long been recognized primarily as a disease of memory. Thus, memory loss has been assumed to be the most sensitive and specific clinical marker of the underlying AD pathology. Recent studies, however, have raised questions as to memory loss’ specificity to AD [17,18,19,20,21,22]. Bertoux et al. [18], in particular, tested the memory of 91 volunteer patients who were in the early stages of cognitive decline. The researchers followed these patients until death after which autopsies were performed. Autopsy results revealed that a third of the patients with AD pathology had no memory losses and half without AD pathology had memory losses. These findings led the researchers to declare that “AD clinical diagnosis cannot rely on the memory profile or severity of amnesia” (p. 11).

In addition to core memory deficits, visual impairments are also pervasive in AD. Among the deficits documented in the literature are visual acuity, contrast sensitivity, color discrimination, backward masking, figure-ground separation, object and face perception, optic flow perception, structure from motion (SFM), stereoacuity, and visuospatial disorientation, [23,24,25,26,27]. These visual deficits will, in turn, severely limit patients’ visuospatial processing capacity, an essential element of daily functioning and fundamental to their maintaining independent living [28].

Broadly defined, visuospatial function refers to, “the ability to identify, integrate, and analyze space and visual form, details, structure, and spatial relations in several (usually two or three) dimensions” ([29], p. 467). Thus, visuospatial dysfunction may underlie AD patients’ tendency to become easily disoriented and get lost—even in the most familiar environments, such as their own homes or neighborhoods [30,31,32,33,34,35,36]. As the disease progresses, incidents of getting lost are likely to increase, leading to increased dependence and decreased autonomy.

Given the prevalence of visual symptoms in AD, researchers have begun exploring the visual system as a potential source of AD biomarkers. Current research has confirmed the presence of AD pathologies in the visual pathways, not only at the subcortical level, but also at the cortical level. However, the visual deficits found in AD appear to occur, not due to pathological changes in the visual pathways up to the primary visual cortex (i.e., at the subcortical level), but due to alterations at the higher cortical level—particularly in the visual association areas [27,37,38,39].

The cortical processing of visual information begins in the primary visual cortex (also known as the striate visual cortex or V1). The information then travels to the secondary visual cortex (also known as V2), where two anatomically distinct and functionally specialized streams of projections emerge carrying information to the surrounding visual association areas (V2, V3). Referred to as “the two visual systems hypothesis”, the hypothesis contends that the “what” pathway projecting to the inferior temporal cortex is specialized for visual object perception and recognition; whereas the dorsal “where” pathway projecting to the posterior parietal cortex is associated with spatial vision and guidance of action [40,41].

The dorsal stream coursing through visual areas V3, V3A, V5/MT+, V6, and V7 and terminating in the posterior parietal cortex has been found to be particularly vulnerable to AD. Specifically, visuospatial deficits are thought to be caused by the presence of pathological lesions of AD in the visual association areas [37,42]. An autopsy study of 41 elderly brain donors revealed a high density of core pathological lesions of AD in the visual association cortices of all subjects known to have had mild cognitive impairment (MCI) and AD [12]. Significantly, the authors also observed that 52% (13 of the 25) of cognitively intact subjects had similar alterations in the visual association cortex.

The dorsal pathway terminates in the posterior parietal lobe. A significant portion of the parietal cortex consists of multimodal association areas wherein somatosensory (the sense of touch and limb position), vestibular, and visual information converge and are integrated with motor signals to control movement and coordinate navigational skills. Bruner and Jacobs [43] surmise that the multimodal nature of information processing and a high synaptic complexity makes the parietal cortex an early target of neuropathological changes. Accumulation of pathological lesions in the grey matter eventually leads to neuronal death and cortical thinning. In a study investigating grey matter volume changes in the parietal lobe, Jacobs et al. [44] observed that the preclinical AD group exhibited a pattern different from those exhibited by the cognitively stable and cognitively declining groups. In addition, the brain is a high energy consumer and relies exclusively on glucose as its sole source of fuel. For those who develop insulin resistance, a pathological condition in which cells fail to respond to insulin effectively, excess glucose builds up in the bloodstream. A growing number of researchers suspect that defects in glucose metabolism trigger a chain of pathological events leading to cognitive impairment and neurodegeneration, resulting in AD [45,46,47]. Based on the review of imaging studies in early AD patients, Jacobs et al. [48] have concluded that metabolic alterations are most prevalent in posterior parietal areas.

Whether it is abnormal protein aggregates, metabolic defects, cortical atrophy, or a combination of these, damage to the dorsal pathway will reduce visuospatial capacity. More importantly, the alterations in the dorsal pathways appear to precede those in the medial temporal lobe, the area known to be responsible for episodic memory [49]. In fact, memory loss, the cardinal symptom of AD, is also prevalent in other types of dementia, such as frontotemporal dementia (FTD) [50,51]. Even the normal aging process takes a toll on memory, rendering memory impairments an inadequate biomarker for AD [20,21,22,37]. To identify an effective measure that can differentiate AD from other neurodegenerative diseases, Yew et al. [52] had patients with AD and behavioral variant FTD perform memory and spatial orientation tests. The authors observed that both patient groups performed poorly on memory tests with the AD group’s performance degrading even further. By contrast, FTD patients demonstrated largely intact capacity on spatial orientation tests with performance comparable to that of healthy controls, whereas the AD group performed poorly. The results suggest that AD and FTD can be discriminated based on a combination of simple memory and spatial orientation measures. Ritchie et al. [53] conducted a study examining cognitive processing in a middle-aged cohort (40–59 years) at risk of developing late onset dementia in an attempt to identify a diagnostic measure to predict AD in the preclinical stage. Based on study results, the authors concluded that spatial processing is a potential indicator of AD—even in the preclinical stages. Taken together, these studies provide convincing evidence for the utility of spatial processing deficits as an early cognitive marker of AD-related pathological alterations (see also [20,21,22]).

### 1.3. Stereopsis in AD

For the sake of illustration, let us define a coordinate system with the fovea of each retina as the origin. If we fixate on an object, its image forms on the origin of each coordinate system. Other objects, however, will cast their images away from the origin. Of these, there will be objects whose images form on the same location of both coordinate systems. These locations are referred to as corresponding points. Assuming that the eye is a sphere, connecting all the objects forming corresponding points defines a circle referred to as horopter. By contrast, those objects not on the horopter project to noncorresponding retinal locations. For any object not on the horopter, a visual angle can be defined for each eye subtended by the object and the fixated object with respect to the nodal point of each eye. Absolute disparity, the difference between the two visual angles, conveys distance information between the object and the observer in relation to the fixated object, that is, nearer or farther than the fixated object. For any two objects lying at different distances from an observer, their relative distance from the observer is conveyed through relative disparity, the difference in absolute disparities of the two objects. In this way, the observer can be aware of his or her distance to an object using absolute and/or relative disparities.

The neural process that extracts the relative depth information of objects with respect to the fixation point using binocular disparity is referred to as stereopsis. Importantly, of many spatial cues that convey depth information, the most potent source may be binocular disparity, which has the capacity to estimate depth intervals far superior to any monocular cues—even as much as by a factor of 40 or better [54,55]. Mendez et al. [37] suspected disturbances in stereopsis as a primary contributor to visuospatial deficits in AD. In fact, a number of studies have corroborated this suspicion with the finding that stereoscopic abilities decline in patients with AD [31,56,57,58,59]. However, some authors reported conflicting evidence of largely intact stereoscopic capacity in patients with AD [27,40]. These conflicting findings need to be resolved if visuospatial dysfunction is to qualify as a biomarker for AD. The present study aims to shed some light on the conflicting findings involving stereopsis and AD. The results should facilitate the development and validation of measures to assess visuospatial dysfunction in clinical trials and diagnosis of disease.

To that end, it is important to recognize that the risk of AD increases with age. Thus, if a decline in stereopsis is observed in patients with AD, it is important to determine whether it is a normal age-related decline or another manifestation of AD. An overview of the literature that has investigated aging and stereopsis revealed a moderate or strong level of performance degradation when assessed in terms of stereoacuity [60,61,62,63]. Norman and colleagues [64,65,66] noted that a majority of the studies directed at aging and stereopsis assessed stereoacuity, i.e., the smallest depth difference that can be detected using binocular disparity. Note that stereo thresholds assessed by the standard random dot stereoacuity test range from 20 to 400 arc sec. However, in the real world human actors interact with an environment comprised of 3-D objects whose perception requires detection of disparities much larger than those of the stereo images used in common tests of stereovision (e.g., TNO, Titmus, Randot, Frisby, etc.) [67]. Recognizing this difference, Norman and colleagues [64,65,66] conducted a series of studies assessing younger and older participants’ stereoscopic abilities to perceive depth intervals and discriminate 3-D surface shapes both with high magnitudes of binocular disparity. Norman et al. [65] observed that older participants perceived about 20% less depth than younger participants. Despite a slight degradation in accuracy, however, older participants exhibited a response pattern like that of younger participants in discriminating depth and 3-D shape, suggesting that their stereoscopic vision retains a substantial amount of functionality, capable of subserving activities of daily living in ordinary environments. As Wilcox and Allison [67] note, most objects comprising the visual world lie outside of Panum’s fusional area, requiring the stereoscopic system to detect large (coarse) disparities for their perception.

Researchers investigating AD and stereopsis also have employed conventional stereoacuity tests, such as the Titmus test [27,31], or the Randot test [37,59]. One exception is a study in which participants watched a 3-D TV drama while wearing a pair of 3-D glasses [58]. Upon completion of the drama, participants were asked to rate the impression of 3-D on a scale of 1 to 5 with 1 being unable to perceive stereoscopic depth and 5 being experiencing a compelling sense of solidity. Of the three groups participating in the study, the AD group scored lowest with an average score of three, followed by the normal control group with an average score of four, followed by the Parkinson’s group with an average score of five. The authors also administered the Titmus test which, however, failed to differentiate the three groups. These findings demonstrated that stereopsis is impaired in patients with AD and led the authors to contend that the 3-D TV test can be more effective than common stereoacuity tests in assessing stereopsis.

To date, as Norman and his colleagues [64,65,66] attest, only a few studies have assessed the stereoscopic visual system’s capacity to perceive depth intervals. Depth intervals, ubiquitous in cluttered environments, require the detection of large disparities to be perceived. In this regard, Lee et al.’s [58] use of a 3-D TV drama to assess stereopsis is reasonable. However, the study employed subjective ratings to assess stereoscopic capacity. An objective assessment would have been preferable with systematic manipulation of the amount, type (absolute vs. relative), and polarity (crossed vs. uncrossed) of disparities associated with the characters and the scenes appearing in the drama.

### 1.4. The Present Study

Stereopsis, the only mechanism capable of conveying a compelling impression of solidity or three dimensionality, plays a prominent role in extracting depth information from the environment [55]. To date, relatively few studies have investigated the effects of AD on stereopsis; and those studies have yielded mixed results. Cronin-Golomb [68] suspects that different demand characteristics associated with each stereoacuity test may have contributed to the conflicting results. It is also important to recognize that, as Landers and Cormack [69] note, poor performance on stereoacuity tests may have to do with the naivety of participants unaccustomed to making depth judgments using binocular disparity as the sole source of distance information. Indeed, it has been repeatedly demonstrated that stereoacuity improves with practice [55,69,70]. Interestingly, McKee and Taylor [55] found that the practice effect associated with stereoacuity tests disappears when assessed with real objects. Based on their findings, the authors expressed caution in evaluating the performance of unpracticed participants on stereoacuity tests.

More importantly, to perceive the visual world with two eyes, the stereoscopic system should be able to detect disparities specifying various depth intervals separating the objects [64,67]. As the primary means of extracting depth information from the environment, deficits in this capacity would severely compromise the overall visuospatial function by making it unable to subserve routine activities, such as reaching, grasping, playing sports, driving, and navigation. In the present study, we examined the stereoscopic abilities of patients with AD and MCI to discriminate various depth intervals separating two virtual objects.

Patients with MCI were included in the study. MCI denotes “a group of individuals who have some cognitive impairment but of insufficient severity to constitute dementia” [71], thus representing a transitional stage between normal aging and dementia. However, with the annual conversion rate of 10–15%, patients with MCI are at high risk of progressing to AD [72]. Despite a significant risk of later development of AD by MCI patients, there are currently no recommended diagnostic criteria to confirm MCI. Instead, MCI is largely diagnosed by clinicians’ judgment based on the results of various neuropsychological tests complemented by lab tests and imaging data. Recently, evidence has been accumulating indicating that visuospatial function declines in MCI as it does in AD [32,73,74,75,76]. In the absence of reliable biomarkers for MCI, it is hoped that if visuospatial dysfunction is specific to AD, it might be also specific to MCI, thereby providing a potential cognitive marker for MCI. For this reason, we examined the stereoscopic capabilities of MCI patients in the present study.

Because participants were made up of older adults and patients with dementia, we simplified experimental procedures to facilitate participants’ cooperation and completion of the task. Specifically, we employed a computer-based, two-alternative, forced choice task that eliminated the need to respond by naming objects or discriminating complex figures. In addition, the experimenter controlled the computer application that presented the stimuli and recorded participants’ responses.

Participants wore red-blue anaglyph glasses to view stereograms depicting two cubes, one left and the other right of the center of the monitor, which appeared at varying distances from the participants. Their relative distances were controlled in two disparity (absolute vs. relative) conditions combined with two disparity (crossed vs. uncrossed) directions. In the absolute disparity condition, one object appeared to lie directly on the computer screen and the other object appeared to be floating in front (crossed disparity) or behind (uncrossed disparity) the computer screen. In the relative disparity condition, both objects appeared in front of the screen (crossed disparity) or behind the screen (uncrossed disparity).

Participants watched virtual images of two cubes under stereoscopic viewing conditions and identified the object that appeared closer to them. Each monocular half-image of the stereogram lacked depth information. This task can only be accomplished when the stereograms are viewed in 3-D after proper fusing of the two half images.

## 2. Materials and Methods

Participants. Twenty-one AD patients (6 males and 15 females), 23 MCI patients (7 males and 16 females), and 21 healthy elderly control (EC) participants (14 males and 7 females) participated in the study. AD and MCI patients, all enrolled in Kyungpook National University Hospital’s outpatient clinic, volunteered for the experiment.

AD patients were selected on the basis of the diagnostic guidelines of the National Institute of Neurological and Communicative Disorders and Stroke-Alzheimer’s Disease and Related Disorders Association (NINCDS-ADRDA) for probable or possible AD [77]. MCI patients were selected on the basis of the diagnostic guidelines of Petersen criteria for MCI [71]. Additional evaluations included neurological examinations, laboratory blood tests, and either CT or MRI scan to exclude other causes of dementia.

Dementia severity was assessed by the Korean adaptation [78] of the Mini Mental State Examination (K-MMSE) [79] and the Clinical Dementia Rating (CDR) Scale [80]. For the AD patients, the mean K-MMSE score was 21.5 (SD = 4.1); and all had a CDR score of 0.5 or 1 (mean CDR = 0.98, SD = 0.10). For the MCI patients, the mean K-MMSE score was 25.4 (SD = 2.8). Elderly controls (EC) were comprised of relatives of patients and temporary workers from the first author’s University Maintenance Department, who received a nominal fee for their participation. AD (mean age = 71.6 years, SD = 9.7; mean education = 6.7 years, SD = 4.4 years), MCI (mean age = 72.8 years, SD = 7.9; mean education = 9.2 years, SD = 4.2 years) and EC (mean age = 69.5 years, SD = 7.8; mean education = 9.2 years, SD = 3.7 years; mean K-MMSE = 28.4, SD = 1.7) groups were matched for age, *F*(2, 62) < 1.0, *ns*, and years of education, *F*(2, 62) = 2.66, *p* > 0.05, but not for MMSE which differed significantly among the three groups, *F*(2, 62) = 27.12, *p* < 0.001. Demographic data for the participants are presented in Table 1.

All participants had normal or corrected-to-normal vision and reported no history of ophthalmologic disorder. Prior to the experiment, we assessed stereoacuity using the Multi-Target Red/Green Anaglyph Stereo Test (Random Dot Butterfly, Letter “E”, and Figures; Synthetic Optics Inc., Franklin Lakes, NJ, USA). The results were inconsistent, with some participants (4 EC, 5 AD, 2 MCI) unable to identify stereo targets with any disparities or some failing to identify targets with large disparities but identifying targets with smaller disparities. Stereoblindness, the inability to perceive depth based on binocular disparity, is actually quite prevalent in the normal population. Although estimates vary depending on experimental methodology, task, and apparatus, it is thought that 5–20% of population is stereo blind or stereo impaired [81]. More significant in the present context is research showing that normal aging is detrimental to stereopsis. Wright and Wormald [63], in particular, administered a Frisby stereo test to 728 elderly adults over the age of 65 and observed that only 27% had full stereopsis but 29% had none, a finding comparable to the present result.

However, as noted earlier, performance on stereoacuity tests improves with practice [59,69,70]. Note that the elderly population in Korea has had limited exposure to 3-D movies or stereoscopic viewers. The participants in the present study, likewise, were inexperienced at perceiving solidity and depth based on binocular disparity, which may have contributed to the inconsistent performance we observed. Indeed, when we inquired of participants about their capacity to perform everyday visual tasks, none expressed difficulties. Wilcox and Allison [67] remind us that, whereas stereoscopic research in the past 50 years concentrated on precision, most of the objects populating our visual world lie outside Panum’s fusional area, thus are diplopic. By detecting coarse disparities, the observer can gain an awareness of 3-D layout of the surrounding environment, which then serves as the perceptual basis for natural tasks such as reaching and grasping, picking up and moving objects, driving, navigation, and playing sports. Norman et al. [64,65,66] demonstrated convincingly that the stereoscopic abilities of older observers in perceiving depth and 3-D shapes with large disparities remain relatively intact. Given these findings, in the current study we assessed stereoscopic processing of relatively large disparities specifying depth intervals of different magnitudes.

Apparatus. The stereograms were generated by a laptop (LG15U56, LG Electronics, Seoul, Korea) equipped with GeForce 940 M, a mobile graphics chip by NVIDIA (NVIDIA, Santa Clara, CA, USA), and displayed on a 23-in LCD monitor with a pixel resolution of 1920 H × 1080 V and frame rate of 60 Hz. Participants viewed the stereograms at a distance of approximately 60 cm. No physical constraints on head movement were imposed during the experiment.

Stimuli. Displays depicted two virtual cubes on a white background (Figure 1). The two cubes, each with a side length of 2.5 cm, appeared to float at eye level, but were displaced laterally 6 cm, one to the right and the other to the left, from the center of the display screen. Each side of the cube was rendered with a different texture. The six texture images were randomized in each trial to produce different images of the cube across trials to eliminate effects of familiarity.

The two cubes appeared at different distances from the observation point with their distances controlled in two disparity types (absolute vs. relative) combined with two disparity directions (crossed vs. uncrossed).

The stereo image pairs were presented simultaneously as anaglyphs. Participants viewed the stereograms wearing glasses with red/blue filters. The stereo images were calibrated in accordance with each participant’s interocular distance.

Design. The experiment was run in two blocks with the first block assessing absolute disparity and the second block assessing relative disparity. In the absolute disparity block, one object appeared on the screen where fixation was held, and the other object appeared either in front of the screen (crossed disparity), or behind the screen (uncrossed disparity). Relative distances (cm) of the two cubes with respect to the screen varied among the pairs of (0.0, 2.0), (0.0, 3.0), (0.0, 4.0), (0.0, 5.0), (−2.0, 0.0), (−3.0, 0.0), (−4.0, 0.0), and (−5.0, 0.0) with the positive values corresponding to uncrossed disparity conditions and the negative values to crossed disparity conditions. Expressed as distances to the observation point, these distance pairs corresponded to (60.0, 62.0), (60.0, 63.0), (60.0, 64.0), (60.0, 65.0), (58.0, 60.0), (57.0, 60.0), (56.0, 60.0), and (55.0, 60.0), respectively. These pairs of values yielded retinal disparities of 12.0, 21.6, 31.2, 42.0, 10.8, 18.6, 25.8, and 33.0 arc min, respectively, (based on the average interocular distance of 6.0 cm).

In the relative disparity block, both objects either appeared in front (crossed disparity) or behind (uncrossed disparity) the screen. Relative distances (cm) of the two cubes with respect to the screen varied among the pairs of (2.0, 4.0), (3.0, 6.0), (4.0, 8.0), (2.5, 7.5), (−2.0, −4.0), (−3.0, −6.0), (−4.0, −8.0), and (−2.5, −7.5). In terms of distances to the observation point, these pairs of values (cm) corresponded to (62.0, 64.0), (63.0, 66.0), (64.0, 68.0), (62.5, 67.5), (56.0, 58.0), (54.0, 57.0), (52.0, 56.0), and (52.5, 57.5), respectively, which in turn corresponded to retinal disparities of 10.3, 14.8, 18.8, 24.2, 12.6,19.9, 28.0, and 33.8 arc min, respectively.

Note that the 4 distance pairs in each condition of disparity type × disparity direction may not be identical to each other in terms of disparity magnitude. However, irrespective of disparity type and direction, they corresponded to 2, 3, 4, and 5 cm, respectively, in terms of linear separation of the two cubes along the depth dimension. On this ground, they are treated as a single variable (depth interval) with 4 levels.

For each distance pair, the first value corresponded to the left cube and the second value to the right cube. The order of these two values was randomized such that either the left cube or the right cube appeared closer to the observer depending on the trial.

These manipulations yielded a 2 (disparity direction: crossed vs. uncrossed) × 4 (depth interval) × 2 (distance order: left vs. right) design for a total of 16 completely randomized trials for each block of disparity type.

Procedure. After completing the informed consent procedure, participants were evaluated with the K-MMSE and stereoacuity tests. The experimenter then measured interocular distance from each participant which was used to calibrate stereo images for the experiment.

The experimenter explained the task to the participant who sat facing the center of the monitor wearing red-blue anaglyph glasses. Each trial began when a small cross appeared at the center of the monitor. With the participant fixated on the marker, the experimenter initiated each trial by triggering two cubes to appear. At that point, the participant was instructed to identify the object that appeared closer to him or her. Participants were encouraged to take as much time as they needed to make a decision.

To familiarize participants with the task, an application was constructed depicting two virtual cubes as described above. Using the application, the experimenter manipulated one of the cubes to slide along the depth axis while the other cube remained stationary. As the cube moved toward, or away from, the viewer, the experimenter made sure that the viewer understood the depth order of the two cubes. Participants then were given an 8-trial practice session prior to the experiment to allow them to become familiar with the experimental setup. Four distance pairs (0, −3.0), (0., 3.0), (−1.0, 4.0), (1.0, 4.0) were combined with 2 distance order conditions to produce the eight practice trials. Feedback was provided during the practice trials, but not during the experiment.

## 3. Results

Overall performance of the three groups is shown as a function of depth interval for each disparity direction in Figure 2. As shown in the figures, all three groups performed adequately with MCI patients performing better with 90% accuracy, followed by AD patients with 87% accuracy, and EC with 84% accuracy. However, these differences in performance did not reach statistical significance, *F*(2, 62) = 1.94, *p* > 0.05. For a detailed analysis, responses were collapsed over distance order and converted into percent correct. The results were entered into a mixed-design analysis of variance (ANOVA) with group as a between-subjects variable and disparity type, disparity direction, and depth interval as within-subjects variables.

The ANOVA revealed main effects of disparity direction, *F*(1, 62) = 8.91, *p* < 0.01, η_p_^2^ = 0.13, and depth interval, *F*(3, 186) = 10.27, *p* < 0.001, η_p_^2^ = 0.14. Except for these two main effects, the ANOVA failed to confirm other main effects or interactions. All three groups performed better in the crossed disparity condition (*M* = 89.6%, *SD* = 11.9) than in the uncrossed disparity condition (*M* = 84.9%, *SD* = 13.4) (see Figure 2). This result replicates previous findings demonstrating superiority of stereoscopic processing of crossed disparities over that of uncrossed disparities [66,69,82,83,84]. Accuracy increased in proportion to the magnitude of separation between the two cubes, a result consistent with the finding of McKee and Taylor [55]. Put simply, participants discriminated large depth intervals more accurately with one exception. The EC group performed poorly, particularly in the uncrossed disparity condition (top panel of Figure 2). Although the source of this response pattern is unclear, it is important to note that no interaction was observed involving either group or disparity direction. It is possible that small sample size may have contributed to this response pattern. We further discuss issues involving sample size in the Discussion section.

Recall that the stereoacuity test administered prior to the experiment yielded inconsistent results. In particular, eleven participants (4 EC, 5AD, 2 MCI) were unable to identify stereo targets with any disparities. To determine whether these participants were truly stereoblind, we assessed their performance separately. The results demonstrated that these participants performed adequately in the present experiment with a mean accuracy of 83%, well above chance (50%), *t*(10) = 9.35, *p* < 0.001. These results suggest that these individuals may be considered stereoblind in terms of stereoacuity, but stereo-capable in terms of suprathreshold (or coarse) disparities.

## 4. Discussion

Our current understanding of AD is that the pathological processes leading to dementia begin long before clinical symptoms emerge. As no effective treatment is available to reverse or prevent the progression of the disease, the research focus has been to identify the disease in its earliest stages in the hope of intervening in the disease process before brain damage has become irreversible. Although recently developed CSF and neuroimaging biomarkers show promising results, their high cost and invasiveness, as well as a lack of adequate medical facilities have prevented their availability to patients with dementia. Thus, there is an urgent need to develop noninvasive and cost-effective diagnostic biomarkers for early detection and accurate diagnosis of AD in the preclinical stages.

Memory loss, the cardinal symptom of AD, is also prevalent in other brain diseases, such as FTD, albeit with different underlying pathologies, while also a part of the normal aging process. Because of overlapping symptoms, memory impairment has proved to be unable to reliably distinguish AD from non-AD dementias [18,20,21,22,50,51]. Recently, visuospatial dysfunction has emerged as a potential marker that can detect AD—even in its earliest stages. The structures comprising the dorsal stream linking V1 to the posterior parietal cortex constitute the core network for spatial vision and guidance of action [42,43]. Whether due to the presence of AD pathology, metabolic decline, cortical atrophy, or a combination of these, a number of studies have indicated that these visuospatial structures are damaged even in the preclinical stages of the disease.

Visuospatial function is a complex function consisting of visual perception, construction, and visual memory [21,22,85]. Of these, the present study focused on deficits in visual perception in AD. Mendez et al. [37] conjectured that abnormal depth perception, caused in part by disturbance in stereopsis, is the prime contributor to visuospatial deficits in AD. To date, few studies have investigated whether AD also impacts stereoscopic abilities, and the results of these studies have been inconclusive [27,31,37,38,56,57,58,59]. Differences in experimental methodology, task, and apparatus, in conjunction with varied sample size and heterogeneity in patient samples, may have contributed to the discrepant results.

The majority of these studies assessed stereopsis using stereoacuity, a threshold measure defined as the smallest detectable depth difference based on binocular disparity. However, for natural tasks such as reaching, grasping, driving and navigation, as Hartle and Wilcox [86] attest, what matters most is the capacity to detect coarse disparities. Thus, we examined whether stereoscopic capacity to detect coarse disparities specifying various depth intervals is impaired in patients with AD and MCI. To this end, we manipulated disparity type (absolute vs. relative), disparity direction (crossed vs. uncrossed) and disparity magnitude to produce two virtual cubes at two different distances from the observer. Using a two-alternative, forced choice task, participants judged the object that appeared closer to them.

As we discuss the present findings, we urge caution in interpreting the implications of the findings. First, COVID-19 disrupted data collection, limiting our sample size. Second, the three groups were matched in terms of age and education, but not in terms of gender. Consequently, males were overrepresented in the EC group (67%) but underrepresented in the two patient groups (29% for AD and 30% for MCI, respectively) with the difference reaching statistical significance, χ^2^(2, *n* = 65) = 8.08, *p* < 0.05. Thus, the gender mismatch among the groups may be a potential source of bias. Although this possibility raises a concern, previous findings demonstrating negligible influence of gender on stereoacuity are somewhat assuring [87]. However, empirical data will be needed to validate whether that is also true with AD patients.

The experimental design was simplified substantially in consideration of participants’ age and cognitive capacity. Nevertheless, it should be underscored that this task can be accomplished only when the two monocular half images are fused properly. As is clear from Figure 1, it is impossible to determine which cube is nearer the viewer from each half image alone (middle and bottom panels of Figure 1). Note that the range of disparities assessed in the present study fell on the coarse scale of 20 to 80 arc min [86,88].

Results were straightforward. Participants were equally accurate in judging depth intervals in the absolute disparity condition (M = 88%, SD = 11.8%) and in the relative disparity condition (M = 87%, SD = 12.1%). More importantly, participants discriminated depth intervals in the crossed disparity condition more accurately than those in the uncrossed disparity condition, replicating a finding demonstrated repeatedly in the literature [66,69,82,83,84]. The fact that the asymmetry in disparity processing, which Landers and Cormack [69] suspect to be a general phenomenon in stereopsis, was observed in all three groups provides strong evidence that stereoscopic abilities are largely preserved—not only in AD patients, but also in MCI patients. The finding that all three groups discriminated larger depth gaps more accurately than smaller gaps, that is, performance improved in proportion to the magnitude of depth interval, further reinforces this observation that the stereoscopic visual system of all three participant groups is functional.

To summarize, the finding that EC performed adequately replicates similar findings reported by Norman and colleagues demonstrating that aging plays an insignificant role on stereopsis [64,65,66]. Significantly, the finding demonstrating that AD and MCI patient groups performed as well as, or even better than EC, presents clear evidence that patients’ coarse disparity processing system is intact, capable of extracting adequate depth information that can be used to subserve daily activities.

Since Wheatstone’s discovery of stereopsis in 1838, numerous studies have been conducted to uncover the neural substrates subserving stereopsis. Current understanding is that, upon receiving signals from each retina, disparity-selective neurons in V1 perform the preprocessing stage and then send the outputs to higher order or associative visual areas (e.g., V2, V3, V3A, VP, MT, MST, and IT) for further processing of depth information. Of the extrastriate visual areas, MT (middle temporal area) has been shown to play a significant role in processing disparity signals [89,90,91,92]. Significantly, and particularly relevant to the present study, it is large (or coarse) disparities to which disparity-selective neurons in MT are sensitive. As it happens, MT is the most extensively researched extrastriate area for its role in higher-order motion processing, e.g., structure-from-motion (SFM) or optic flow, both of which work together to subserve spatial navigation. Significantly, it has been demonstrated repeatedly that perceptual capacities to process these two types of motion signals are severely compromised by AD with deficits in SFM processing [27,57,58,93,94,95,96,97] and in optic flow perception [32,98,99,100,101,102].

These findings, taken together with those of the current study, suggest that, of the MT neurons with different functional properties, disparity-selective neurons are largely intact and unaffected by AD and therefore functional; whereas optic flow and SFM-sensitive neurons are severely damaged by AD and therefore dysfunctional. Interestingly, Thiyagesh et al. [38] observed similar results in an fMRI study. To locate the neural basis of visuospatial deficits in AD, the authors administered fMRI to AD patients and healthy controls while participants viewed a radially expanding optic flow pattern and stereo motion. The authors observed that the patient group revealed no evidence of impaired stereopsis. However, whereas the control group demonstrated activation of MT in response to stereo motion stimuli, AD patients failed to activate MT. Instead, the authors observed that AD patients activated the inferior parietal lobule, an area thought to be involved in motion processing [103,104] but not known to respond to binocular disparity, and some additional areas. Moreover, only the patient group showed activation in this region. The authors contended that their findings serve as corroborating evidence for "degenerate systems theory"; that is, degeneracy in a task-subserving region leading to the recruitment of alternate structures to maintain functional integrity [105,106]. In the absence of neuroimaging data, we can only speculate whether the preserved stereoscopic capacity of AD and MCI patients we observed provides further evidence corroborating Thiyagesh et al.’s construal. We must leave this issue for future research.

## 5. Conclusions

Although rather neglected, visuospatial deficits have long been recognized as a common feature of AD. More significantly, recent neurophysiological and imaging studies have confirmed the presence of visuospatial deficits even in the early stages of AD. The ability to perceive accurately objects and their spatial relationships in the environment enables the actor to be successful in daily interactions with the surrounding environment, as in navigating the neighborhood, or reaching for and grasping objects. Impairments in visuospatial abilities, therefore, would have a profound impact on an individual’s daily activities and quality of life.

As underscored above, there is an urgent need to develop minimally invasive and cost-effective biomarkers that can detect AD in its earliest stages, preferably before irreversible brain damage has occurred. Accumulating evidence suggests that memory loss, the hallmark symptom of AD, is not sensitive enough to distinguish AD from other dementias. Visuospatial dysfunction may fill this gap as an important source of AD biomarkers.

Visuospatial function is a complex skill that demands the integration of information from different modalities which, in turn, is combined with motor signals to control and coordinate the movement system for successful interaction with the surrounding environment. If this function is dysfunctional and if this dysfunction is to be exploited as an indicator of a disease, it is essential to have a clear profile of the nature of the dysfunction. In an effort to unpack the nature of the dysfunction, the present study touched on a small component of this complex skill, that is, stereopsis. An experiment directed at the stereoscopic capacity in AD and MCI patients revealed that the two patient groups performed as well as, or even better than EC, clear evidence that patients’ coarse disparity processing system is intact, capable of extracting adequate depth information. At the same time, stereoscopic ability is rejected as a measure that can be utilized to discriminate AD from EC. Nevertheless, it is hoped that the present findings can shed some insight as to the exact cause(s) of visuospatial deficits in AD and also for the development and validation of measures to assess visuospatial dysfunction in clinical trials and diagnosis of disease.

## Figures and Tables

**Figure 1 healthcare-09-00157-f001:**
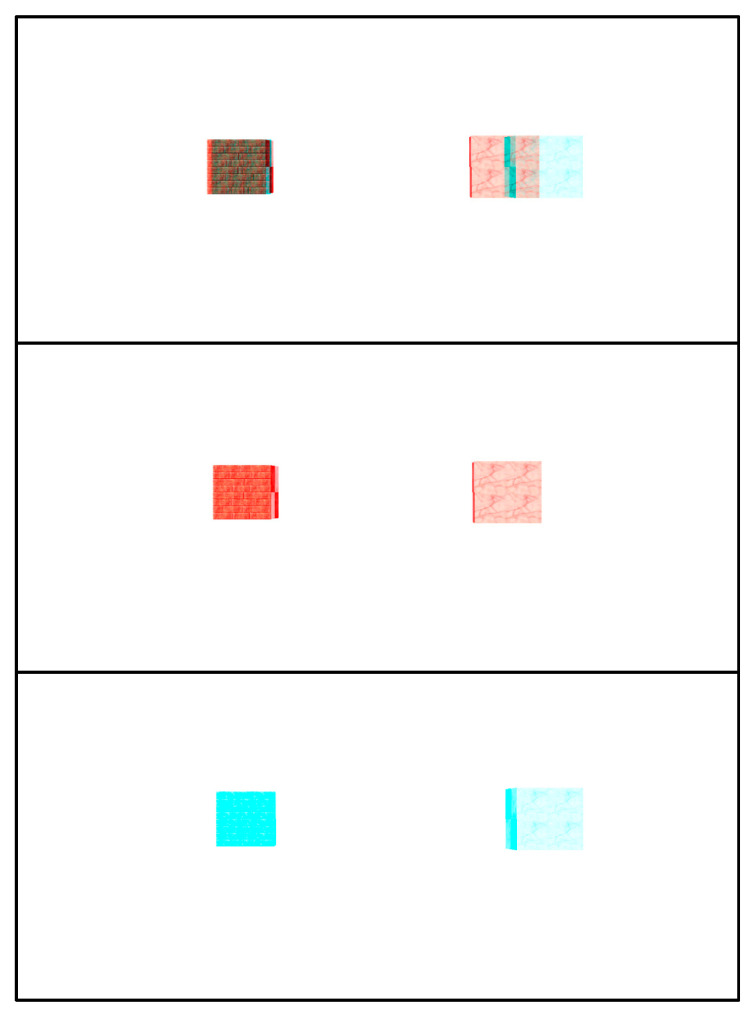
A stereogram used in the present study. (**top**) Two cubes were rendered in cross disparity such that, when viewed with red-blue anaglyph glasses, both appear floating in front of the monitor. The cube on the right appears closer to the observer; (**middle**) half image for the right eye; (**bottom**) half image for the left eye.

**Figure 2 healthcare-09-00157-f002:**
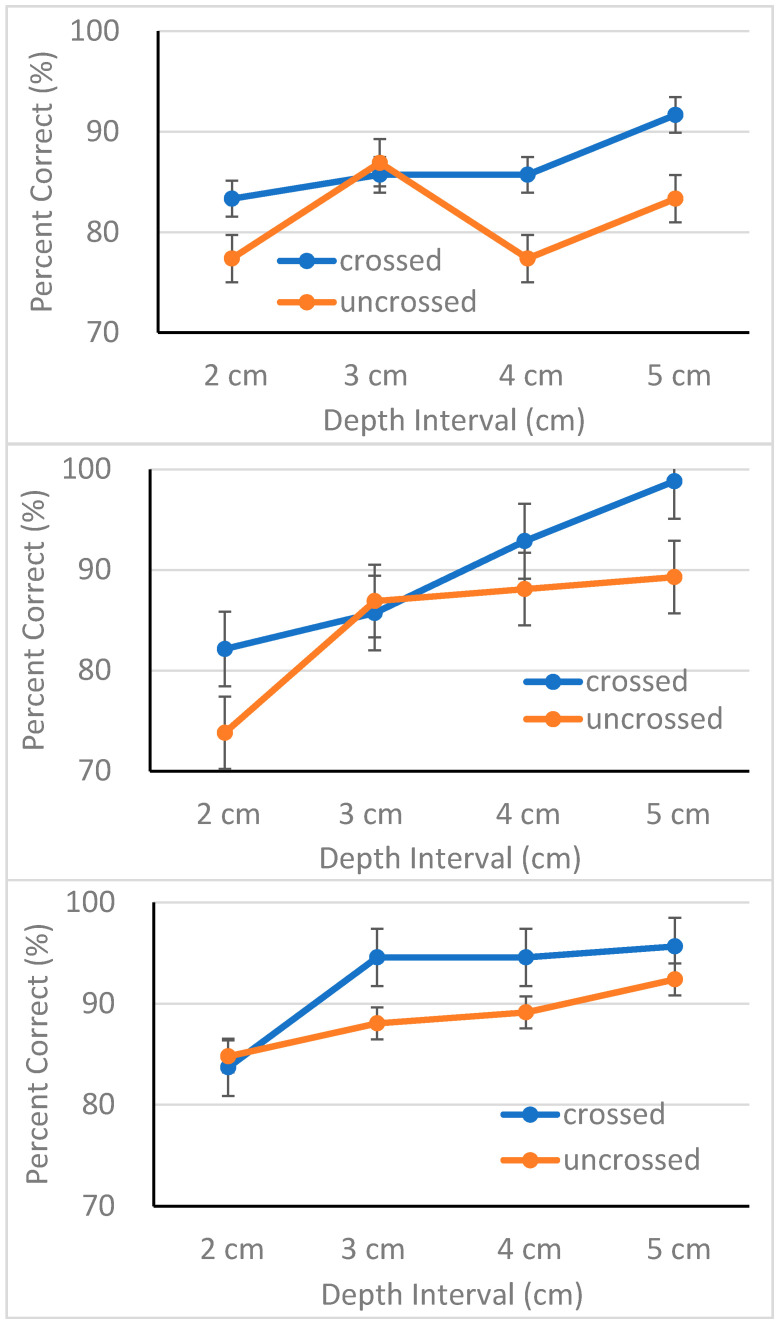
Percent correct (with standard error bars) plotted as a function of depth interval for each condition of disparity direction for elderly controls (**top panel**), AD patients (**middle panel**) and MCI patients (**bottom panel**), respectively.

**Table 1 healthcare-09-00157-t001:** Demographic data of participant groups.

	EC (*n* = 21)	AD (*n* = 21)	MCI (*n* = 23)	*p*-Values
Age (years)	69.5 ± 7.8	71.6 ± 9.7	72.8 ± 7.9	0.43
Edu (years)	9.2 ± 3.7	6.7 ± 4.4	9.2 ± 4.2	0.08
MMSE	28.4 ± 1.7	21.5 ± 4.1	25.4 ± 2.8	0.001

Notes: Data presented as mean ± SD. Abbreviations: EC, elderly controls; AD, Alzheimer’s disease; MCI, mild cognitive impairment; MMSE, Mini Mental State Examination.

## Data Availability

The data presented in this study are available on request from the corresponding author.

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
