# Peer review of "Stereoscopic Depth Perception and Visuospatial Dysfunction in Alzheimer’s Disease"

_healthcare, 2021, doi:10.3390/healthcare9020157_

Round 1

Reviewer 1 Report

This study presents a null result that AD patients perform as controls on a disparity task. The experiment is clearly described and the results are clear as well. However, I think the experimental design and stimulus choice are not properly justified and are likely not precise enough to observe expected changes as stated in the intro. Therefore, the conclusions drawn are much too strong given the specificity of the stimulus and procedure used.

The introduction is not well structured and not clear. The authors want to give too much information and thus the important information is hard to grasp. There are lot of redundancies both within and between the introduction and the discussion. The descriptions of the visual impairments and their neurophysiological substrates are not precise enough. The authors need to explain in a much clearer way the rationale for their experimental design.

All the authors argumentation and discussion revolve around the role of the dorsal pathway and area MT. However, it seems to me that their stimulus is more tuned to the ventral pathway. This needs to be clarified.

The stimuli used are not cubes, they are square shapes.

94-106: lot of redundancies

107-114 not clear. V2 is not the extrastriate cortex. All areas besides V1 are extrastriate

116-124 You speak about the dorsal pathway but then about deficits which involve the ventral pathway (color, form etc..)

You mostly speak about damages to the dorsal pathway (126, 145) but your stimulus is more tuned to the ventral pathway.

Lines 166-171: I think this paragraph is misplaced in the visual function section.

173-189 this part is very confusing (ie: Line 174 “If the observer’s two eyes are superimposed on each other”). By wanting to be too precise the authors make very confusing sentences. I guess they could give simpler definitions of disparity.

263 “MCI was included in the study” I guess you mean MCI patients.

377: “The stereo images were calibrated in accordance with each participant’s interocular distance.” What does this mean?

405: reversed: randomized?

407: only one trial by condition?

412: “The experimenter then measured interocular distance from each participant which was used to calibrate stereo images for the experiment.” If I understand correctly, you did that to adjust the on-screen disparity to consistently give the same depth amplitude to subjects. In that case, the disparity set in arcmin might slightly differ from subject to subject, am I right?

Procedure: Presentation duration?

Fig2: put some grid

428-430: on average for the 4 distances?

Were the stereoblind/stereoanomalous excluded from the analysis ?

One reason you didn’t observe any difference between groups could be that you only used suprathtreshold stimuli. You could have tested relative disparity threshold. Please discuss

I don’t see how your stimulus is tuned to coarse disparities (see line 545 but in the intro too)

553-556 This conclusion is much too strong given your psychophysical results.

572-578 a simpler hypothesis would just be that your stimulus doesn’t target MT.

Author Response

C: 94-106: lot of redundancies

R: As per the reviewer’s advice, a portion of this paragraph (94-101) was deleted.

C: 107-114 not clear. V2 is not the extrastriate cortex. All areas besides V1 are extrastriate.

R: Per the reviewer’s advice, ‘extrastriate cortex’ was deleted from ‘(also known as extrastriate cortex or V2).

C: 116-124 You speak about the dorsal pathway but then about deficits which involve the ventral pathway (color, form etc..)

R: We realized that the phrase “(e.g., inability to discriminate form, color, contrast, difficulties in spatial orientation and motion perception, etc.)” was nothing but confusing. We deleted it in the revision.

C: You mostly speak about damages to the dorsal pathway (126, 145) but your stimulus is more tuned to the ventral pathway.

R: To make it clear that our task is not about perceiving form or color (the function of the ventral pathway), but processing binocular disparities of coarse scale, we added figures demonstrating that the task can be accomplished only when the two half images are fused. Please see similar comments provided below and our responses.

C: Lines 166-171: I think this paragraph is misplaced in the visual function section.

R: As per the reviewer’s advice, this paragraph was deleted.

C: 173-189 this part is very confusing (ie: Line 174 “If the observer’s two eyes are superimposed on each other”). By wanting to be too precise the authors make very confusing sentences. I guess they could give simpler definitions of disparity.

R: Per the reviewer’s advice, we tried to rephrase this part in simpler language (ll. 159-178).

C: 263 “MCI was included in the study” I guess you mean MCI patients.

R: As per the reviewer’s advice, it is replaced by “patients with MCI”.

C: 377: “The stereo images were calibrated in accordance with each participant’s interocular distance.” What does this mean?

R: To draw images on the computer monitor in real scale, we enter all the parameters defining the viewing geometry, such as viewing distance, dimensions of the monitor, and so on. To render images in 3D, we need to add the distance between the observer’s two eyes, which are separated about 50 – 70 mm depending on each individual. This distance is referred to as interocular or interpupillary distance.

C: 405: reversed: randomized?

R: Per advice, “reversed” is replaced by “randomized.”

C: 407: only one trial by condition?

R: Yes, only one trial per condition. But there were 4 different magnitudes (2, 3, 4, 5 cm) of depth intervals for each condition of disparity direction × disparity type.

C: 412: “The experimenter then measured interocular distance from each participant which was used to calibrate stereo images for the experiment.” If I understand correctly, you did that to adjust the on-screen disparity to consistently give the same depth amplitude to subjects. In that case, the disparity set in arcmin might slightly differ from subject to subject, am I right?

R: That’s correct. The disparities corresponding to the depth intervals described in the Design subsection were calculated using an interocular distance of 6.0 cm, the average value of the present participants. To this clear, we added “(based on the average interocular distance of 6.0 cm)” in ll. 405-406.

C: Procedure: Presentation duration?

R: Viewing time was not restricted. To make this explicit, we added “Participants were encouraged to take as much time as they needed to make a decision” in lines 407-408.

C: Fig2: put some grid

R: Per request, gridlines are added to Figure 2.

C: 428-430: on average for the 4 distances?

R: yes, they were averaged values although for each condition of distance there were two trials, one absolute and one relative disparity.

C: Were the stereoblind/stereoanomalous excluded from the analysis ?

R: We added the following paragraph (ll. 517-524) to address this issue:

“Recall that the stereoacuity test administered prior to the experiment yielded inconsistent results. Eleven participants (4 EC, 5AD, and 2 MCI) were unable to identify stereo targets with any disparities. To determine whether these participants were truly stereoblind, we assessed their performance separately. The results demonstrated that these participants performed adequately in the present experiment with a mean accuracy of 83%, well above chance (50%), t(10) = 9.35, p < .001. These results suggest that these individuals may be considered stereoblind in terms of stereoacuity, but stereo-capable in terms of suprathreshold (or coarse) disparities.”

C: One reason you didn’t observe any difference between groups could be that you only used suprathtreshold stimuli. You could have tested relative disparity threshold. Please discuss.

R: As we acknowledged in the submitted ms as well as in the revised ms, COVID-19 forced us to keep the contact with patients short. And then due to participants’ age and cognitive capacity, we had to make the design simple. That is, we could not use strict control and systematicity demanded by a typical psychophysical experiment. We intend to address these issues in the near future.

C: I don’t see how your stimulus is tuned to coarse disparities (see line 545 but in the intro too)

R: To make the stereoscopic nature of the task more explicit, we added each half image of the stereogram to Figure 1. Then we underscored that the task of identifying the nearer cube can be accomplished only when the two half images are fused properly. We further pointed out that the range of coarse disparities as defined by Badcock and Schor (1985; also Wilcox and Allison, 2009) was 20 to 80 arc min, a range conforming to the extent of disparities examined in our study. To that extent, we believe that our task demanded processes involved in coarse disparities and accurate performance by our participants can only be explained by the activation of disparity selective cortical neurons.  

C: 553-556 This conclusion is much too strong given your psychophysical results.

R: We shortened the discussion on the neural basis of stereopsis. What’s left is largely the argument presented by Thiyagesh et al. who found similar results using fMRI.

C: 572-578 a simpler hypothesis would just be that your stimulus doesn’t target MT.

R: As we responded to the similar points commented above, we believe that, given that 1) our task can only be accomplished with the proper fusing of the two half images; 2) the amount of disparities manipulated fell on the coarse disparity range of 20 to 80 arc min; 3) most of our participants performed accurately, we believe that it must have been the results of the activation of disparity selective cortical neurons. We acknowledged however that whether it involved MT neurons was purely speculation in the absence of neuroimaging data (ll. 627-630).

Reviewer 2 Report

In this manuscript, in order to develop easily accessible, cost-effective and non-invasive biomarkers for early detection of AD, Kim, N-G and Lee, H-W have examined the stereoscopic abilities of patients with AD and MCI to discriminate various depth intervals separating two virtual objects. I agree with the authors that in the absence of a non-invasive and cheap diagnostic marker for AD, promising alternatives should be explored. This is even more important given that not many studies have investigated the effects of AD on stereopsis and therefore this manuscript is important. Overall the manuscript is well written

However, the authors should respond and address the following points.

1). Although many types of visuospatial disturbances have been reported in AD patients, whether they occur relatively early in the course of disease so that they can be successfully used as a marker of disease is highly questionable. Also, the presence of high density of core neuropathological lesions in the visual association cortices of both AD and normal subjects further complicates specificity of this marker

2). As authors have indicated in the manuscript, many patients show no memory deficits even with markers of AD pathology and conversely many patients show memory deficits even without AD pathology. This only explains the complex and a highly heterogeneous nature of Alzheimer’s disease and identifying specific marker for AD especially at early stages is not an easy way

3).The following sentence should be reworded, not factual. “the presence of AD pathology in cognitively normal individuals suggests that alterations in the brain may have begun years before clinical symptoms become apparent”.

4). The authors argue that memory impairment alone is not sensitive enough to distinguish AD from other dementias such as FTD. But visuospatial impairment is also not specific to AD, and is well recognized in a wide range of neurological disorders including PD patients with dementia, dementia with Lewy bodies (DLB) and also patients who had recent stroke attack. Thus although visual ad spatial impairment may be sensitive enough to detect, specificity is the greatest obstacle in developing any biomarker for AD.

5). Since two thirds of patients diagnosed with AD are females, the authors should have included more female elderly control (EC) patients similar to MCI and AD groups.

6). How did the authors decide that 8-trial practice session was sufficient to familiarize the experimental procedure.

7). Any explanation why EC group performed poorly in the uncrossed disparity condition?

8). More complex and a wide variety of other visuospatial methods should have been tested to support the conclusions.

Author Response

1). Although many types of visuospatial disturbances have been reported in AD patients, whether they occur relatively early in the course of disease so that they can be successfully used as a marker of disease is highly questionable. Also, the presence of high density of core neuropathological lesions in the visual association cortices of both AD and normal subjects further complicates specificity of this marker.

R: We acknowledge that any issue concerning AD creates controversy, no research finding would be definitive, and thus we must be circumspect. That said, and taking the reviewer’s advice, we have rewritten the Discussion and Conclusion sections underscoring the complex nature of human visuospatial abilities. Recognizing that we are addressing only a small component, that is, stereopsis, we hope that our research findings can still shed some insight as to the effects of AD on this functional capacity.

2). As authors have indicated in the manuscript, many patients show no memory deficits even with markers of AD pathology and conversely many patients show memory deficits even without AD pathology. This only explains the complex and a highly heterogeneous nature of Alzheimer’s disease and identifying specific marker for AD especially at early stages is not an easy way

R: Please see our response to the first comment.

3). The following sentence should be reworded, not factual. “the presence of AD pathology in cognitively normal individuals suggests that alterations in the brain may have begun years before clinical symptoms become apparent”.

R: Per the reviewer’s advice, the phrase is rephrased as “neuropathological processes of AD have begun years before clinical symptoms become apparent.”

4). The authors argue that memory impairment alone is not sensitive enough to distinguish AD from other dementias such as FTD. But visuospatial impairment is also not specific to AD, and is well recognized in a wide range of neurological disorders including PD patients with dementia, dementia with Lewy bodies (DLB) and also patients who had recent stroke attack. Thus although visual ad spatial impairment may be sensitive enough to detect, specificity is the greatest obstacle in developing any biomarker for AD.

R: As we pointed out in our response to the first comment, no research findings involving AD would be free from controversy. Thus, in the revision, we tried to be more focused, listing our objective as “If this function is dysfunctional and if this dysfunction is to be exploited as an indicator of a disease, it is essential to have a clear profile of the nature of the dysfunction. In an effort to unpack the nature of the dysfunction, the present study touched on a small component of this complex skill (ll. 650-652).

5). Since two thirds of patients diagnosed with AD are females, the authors should have included more female elderly control (EC) patients similar to MCI and AD groups.

R: In recognition of this issue, we added the following paragraph (ll. 564-573) in the revision.

“As we discuss the present findings, we urge caution in interpreting the implications of the findings. First, COVID-19 disrupted data collection, limiting our sample size. Second, the three groups were matched in terms of age and education, but not in terms of gender. Consequently, males were overrepresented in the EC group (67%) but underrepresented in the two patient groups (29% for AD and 30% for MCI, respectively) with the difference reaching statistical significance, c2(2, n = 65) = 8.08, p < .05. Thus, the gender mismatch among the groups may be a potential source of bias. Although this possibility raises a concern, previous findings demonstrating negligible influence of gender on stereoacuity are somewhat assuring [87]. However, empirical data will be needed to validate whether that is also true with AD patients.”

6). How did the authors decide that 8-trial practice session was sufficient to familiarize the experimental procedure.

R: We failed to mention a demo application we constructed to familiarize participants with the task. It is added to the revision (ll. 437-443) as follows:

“To familiarize participants with the task, an application was constructed depicting two virtual cubes as described above. Using the application, the experimenter manipulated one of the cubes to slide along the depth axis while the other cube remained stationary. As the cube moved toward, or away from, the viewer, the experimenter made sure that the viewer understood the depth order of the two cubes. Participants then were given an 8-trial practice session prior to the experiment to allow them to become familiar with the experimental setup.”

7). Any explanation why EC group performed poorly in the uncrossed disparity condition?

R: In response, we expanded the paragraph (ll. 513-516) as follows:

“Although the source of this response pattern is unclear, it is important to note that no interaction was observed involving either group or disparity direction. It is possible that small sample size may have contributed to this response pattern. We further discuss issues involving sample size in the Discussion section.”

8). More complex and a wide variety of other visuospatial methods should have been tested to support the conclusions.

R: As in our responses to C1 and C4, we revised the Discussion and Conclusion sections to reflect our narrowly focused objective which was to gain some understanding as to how AD impacts human visuospatial abilities. Because visuospatial skill is a complex function, we could tackle only a small component, that is, stereopsis. With collective effort, we may advance, eventually conquering this disease.